# A Pilot Evaluation of a School-Based Nutrition Education Program with Provision of Fruits and Vegetables to Improve Consumption Among School-Age Children in Palau

**DOI:** 10.3390/nu17060994

**Published:** 2025-03-12

**Authors:** Shu-Fen Liao, Hsu-Min Tseng, Jong-Dar Chen, Chiao-Ming Chen, Sing-Chung Li

**Affiliations:** 1Graduate Institute of Management, Chang Gung University, Taoyuan City 33302, Taiwan; ritaliao1012@gmail.com; 2Department of Education, Shin Kong Wu Ho-Su Memorial Hospital, Taipei 11101, Taiwan; 3Department of Health Care Management, College of Management, Chang Gung University, Taoyuan City 33302, Taiwan; tsenghm@mail.cgu.edu.tw; 4College of Medicine, Fu-Jen Catholic University, New Taipei City 24205, Taiwan; 5Department of Family Medicine and Occupational Medicine, Shin Kong Wu Ho-Su Memorial Hospital, Taipei 11101, Taiwan; 6Department of food Science, Nutrition, and Nutraceutical Biotechnology, Shih Chien University, Taipei 10462, Taiwan; charming@g2.usc.edu.tw; 7School of Nutrition and Health Sciences, College of Nutrition, Taipei Medical University, Taipei 11031, Taiwan

**Keywords:** nutrition education, fruits and vegetables (FVs) consumption, children, nutritional knowledge, attitudes, behaviors

## Abstract

**Background/Objectives**: Children spend the most time in school, which can be a friendly environment that encourages students to eat healthy foods, so school is a good place to learn and practice these behaviors. In Palau, public schools offer free lunches to students. Based on the constructs of Social Cognitive Theory, we conducted a pilot study to evaluate the effects of a school nutrition education program (NE) and the provision of more fruits and vegetables (FV) in school lunches on elementary school students’ nutrition knowledge, as well as their attitudes and behaviors regarding fruit and vegetable consumption. **Methods**: A public elementary school participated in our trial from February to May 2017. While all students (ages 5–12) at the school were provided with free meals through the National School Meal program, only students in grades 4–8 were invited to participate in a four-month nutrition education program and taste-testing activities, during which more fresh fruits and vegetables were incorporated into the school lunches. Participants filled out questionnaires before and after the trial. **Results**: Our results showed that students in grades 4–8 (*n* = 92) had improved scores in all nutrition knowledge items after completing the nutrition education course compared to before the course. Among these, the food category, food servings, and total scores showed statistically significant differences. The scores for attitudes and behaviors regarding FVs intake showed an increasing trend after the completion of the course. Further, the age-based analysis revealed that fifth-grade students showed the greatest improvement in their nutrition knowledge scores, with an increase of 38.5%. However, after adjusting for gender, the significance was no longer observed. Regarding attitude and behavior scores toward fruit and vegetable consumption, fourth-grade students showed the greatest improvement, with increases of 10.9% and 6.3%, respectively. Additionally, we tracked the total amount of FVs consumed by the school annually, noting a consistent rise from 1853 pounds in 2017 to 6580 pounds in 2021, a 3.5-fold increase. **Conclusions**: This pilot study showed that school nutrition education and providing more fresh fruits and vegetables (FVs) can significantly improve children’s nutrition knowledge, while there was only a slight upward trend in attitudes and behaviors toward FV consumption. However, regularly offering nutrition education courses combined with providing more fruits and vegetables in school lunches could effectively increase children’s consumption of these healthy foods.

## 1. Introduction

Childhood is a critical period for lifelong wellbeing, as well-balanced and sufficient nutrients are essential for maintaining good health throughout life [1,2,3]. Conversely, unhealthy dietary habits, such as an excessive intake of saturated fats, trans fats, sugar, and salt, significantly increase the risk of obesity and noncommunicable diseases, including hypertension, cardiovascular diseases, diabetes, and certain cancers [4,5]. These poor dietary choices also contribute to a greater burden on the healthcare system in adulthood [6].

Children spend the most time in school, which can be a friendly environment that encourages students to eat healthy foods and engage in regular physical activity, so school is a good place to learn and practice these behaviors [7]. Therefore, an active component approach includes family members, teachers, community members, and dietitians [8,9]. School Feeding Programs (SFPs) are quite common in developing countries, as providing more fresh fruits, whole grains, and a greater variety of vegetables could lead to more health benefits [10,11]. School-based nutrition education programs have shown success in boosting fruit and vegetable consumption among students, while also enhancing their nutrition knowledge, attitudes, and behaviors [12,13].

The Republic of Palau, made up of 16 states and with a population of about 20,000, is one of the most successful economies among Pacific small island nations. The country faces external economic shocks and climate change, with tourism serving as a key income and employment source. Official statistics indicate that agriculture and forestry contribute only 1.6% of its GDP [14]. Palau has extensive undeveloped land, but attracting local agricultural labor is challenging, leading to limited local agrarian products [15]. In Palau, public schools offer free lunches to students [16]. The Ministry of Education is responsible for preparing monthly menus and centralizing the procurement of school lunch foods from a few primary suppliers for approximately 16 primary schools in Koror and other states. The Ministry of Education distributes these foods to schools every week, and the schools prepare and serve meals according to the lunch menu. A typical menu includes rice, meat, or fish with a small amount of vegetables, canned fruit, and a cup of water [17,18].

Research indicates that noncommunicable diseases (NCD) have become significant health threats in Palau, primarily due to unhealthy diets, lack of physical activity, and overweight [19]. The WHO STEPwise approach to NCD risk factor surveillance (STEPS) for 2011–2013 revealed that 92% of Palau residents aged 25 to 64 did not meet the recommended five daily servings of fruits and vegetables (FVs). Among them, 30.3% of men and 23.3% of women reported consuming no FVs at all [20].

Taiwan Technical Mission (TTM) has planned a farm-to-table project aimed at cultivating seasonal organic FVs for school children and systematically supplying these FVs to schools for free as part of their lunch program. The SKMP (Shin Kong Hospital Medical Assistance Program to Palau) will be responsible for the later-stage work, including promoting children’s awareness of FVs, helping them understand the health benefits of FVs, and encouraging their preference for consuming them [21,22].

Social Cognitive Theory (SCT) has been widely applied in health promotion and disease prevention. When designing interventions based on this theory, the focus should be on enhancing individual self-efficacy, using observational modeling to facilitate learning, altering or optimizing the environment to provide supportive resources that encourage positive behaviors, and strengthening anticipated positive outcomes through positive feedback and reward mechanisms. This theory emphasizes that intervention design must simultaneously consider factors at multiple levels (individual, environmental, and policy levels) in order to promote behavior change [23,24]. In this pilot study, we aim to use the SCT theory to change students’ attitudes and behaviors toward fruit and vegetable intake. We created a supportive social environment through policies that systematically improve school meals by offering more fresh fruits and vegetables. Additionally, we designed a nutrition education curriculum to enhance students’ nutrition knowledge, and conducted cooking and tasting activities to build self-efficacy in trying various fruits and vegetables. It is anticipated that, through an intervention based on SCT, children will gain nutrition knowledge, develop positive attitudes toward fruit and vegetable consumption, adopt healthier dietary behaviors, and ultimately increase their long-term intake of fruits and vegetables.

## 2. Materials and Methods

### 2.1. Study Design and Participants

We conducted a pilot trial to evaluate the effects of a school-based nutrition education program (NE) with the provision of fruits and vegetables (FVs) on elementary school children’s nutrition knowledge, attitudes, and behaviors toward fruit and vegetable intake. Our study included a 16-week nutrition education curriculum and offering more fresh fruits and vegetables in school lunches during the course. Before the program began, a baseline questionnaire was administered to assess children’s nutrition knowledge, attitudes, and behaviors related to fruit and vegetable intake prior to the study. Upon completion of the program, the same questionnaire was used to evaluate whether the nutrition education program and the provision of fruits and vegetables in school lunches changed these aspects.

A public elementary school located in Koror agreed to participate in our trial from Feb to May 2017. First, we communicated with the chefs involved in the school lunch program to discuss the menu and incorporated the free FVs into the meals. The vegetables and fruits included in the school lunch menus and cooking activities during the program were supplied by TTM’s farm, providing organic and fresh produce. These included kingkang, eggplant, napa, cucumber, tomato, carrot, watermelon, banana, and papaya, which were used in both raw and cooked forms. In the school lunches, lower-grade students were provided with an average of 1 serving of vegetables and 0.5–1 serving of fruits, while middle- and upper-grade students were provided with an average of 1–1.5 servings of vegetables and 0.5–1 serving of fruits. All students (aged 5–12 years) in the school receive a free meal from the National School Meal Program. Lower-grade students are provided with 350–450 kilocalories per day, while middle- and upper-grade students are provided with 450–650 kilocalories per day.

Based on the findings of Lippe et al. [25], we developed a curriculum and an outcome evaluation questionnaire to address the nutritional issues of students in Palau. These issues include insufficient consumption of FVs, excessive intake of sugary beverages, and irregular breakfast habits. This study was approved by the Ethics Committee of the Palau Institutional Review Board (PIRB-2014-10). Regarding participant recruitment, only children in grades 4 through 8 were included in the study due to limitations in their writing and reading abilities. Two members of the research team visited each classroom (grades 4–8th) to describe the purpose and procedures of the program. All students were invited to participate in nutrition education classes and activities. However, if both students and parents agreed to fill out the pre- and post-course questionnaires, parents needed to sign an informed consent form before the course began. Participants completed a pre-course questionnaire before the trial began. They then underwent a 4-month nutrition education program and activities, meeting once a week for one hour each lesson. After completing the program, participants filled out a post-course questionnaire.

### 2.2. Nutrition Education Curriculum Design

Based on the Palau Healthy Eating pamphlet and the U.S. Dietary Guidelines, we developed an NE program that included four nutrition topics: healthy eating, food categories, food servings, and healthy snacks. Topic 1 introduced healthy eating by covering the fundamentals of a balanced diet and highlighting the numerous benefits of consuming FVs. This topic also discussed vitamins, minerals, and energy balance, and promoted the use of local foods. Topic 2 covered food categories, focusing on understanding different types of food and planning a balanced breakfast. Topic 3 focused on food servings, helping students understand the appropriate amounts for a healthy meal. Topic 4 covered healthy snacks, including an overview of nutrient labels and a focus on the drawbacks of soft drinks and high-fat foods. Topic 1 was taught over 4 lessons, while topics 2, 3, and 4 were each covered in 3 lessons. To increase classroom engagement and enhance students’ understanding, we employed a variety of methods in addition to traditional lectures, including small games, group discussions, and classroom experiments (Table 1). In addition to the above courses, we also held three FVs cooking classes, where students learned about different FVs and simple preparation methods, and participated in tasting and shared their feedback (Figure 1).

### 2.3. Questionnaire Design

The questionnaire is constructed based on social cognitive theory to investigate the changes in students’ nutritional knowledge, attitudes, and behaviors before and after class. The questionnaire includes two sections. The first part assesses students’ nutritional knowledge and consists of four subsections: (1) Food Categories. This subsection contains 7 questions, e.g., “Which food does NOT belong in the grain group?”; (2) Healthy Foods. This subsection contains 5 questions, e.g., “Which of the following would be a healthy choice for a snack?”; (3) Food Servings. This subsection contains 2 questions, e.g., “How many total cups of fruit and vegetables combined should you eat each day?”; (4) Health Benefits of Breakfast. This subsection contains 3 questions, e.g., “Why is breakfast important?” Students received 1 point for choosing the correct answer and 0 points for choosing an incorrect answer. Questions were adapted from a previously validated instrument [26]. The second section evaluates attitudes and behaviors regarding FVs through 12 trichotomous items (yes/no/maybe). For example: “I feel good when I eat vegetables”. Students receive 3 points for answering “Yes”, 2 points for “Maybe”, and 1 point for “No”. For the Attitudes section, two questions are reverse-coded, so the scoring will be reversed accordingly. Questions were adapted from a previously validated instrument [27] (Appendix A).

### 2.4. Statistical Analysis

We used the nutrition knowledge score as the data for the sample size analysis, setting the mean difference as 2.0, the standard deviation of differences as 4, the alpha two-sided as 0.05, and the power as 0.9. The sample size for a paired *t*-test was calculated to be 44 participants, indicating that the sample size we collected had sufficient statistical power. Data were analyzed by IBM SPSS Statistics 25.0. Continuous variables (knowledge, attitude, and behavior score) were presented as mean and standard deviation (mean ± SD). Since there was a statistically significant difference in gender among the students who agreed to participate in the study, a two-way repeated measures ANOVA was used for pre- and post-test comparisons to control for this difference. A *p*-value < 0.05 is considered statistically significant.

## 3. Results

### 3.1. Increase the Supply of FVs in School Lunches

In the past, school lunches provided fewer fresh FVs, and sometimes included canned FVs (Figure 2A). After discussing with the chef, the school lunch menu was modified to include more fresh fruits and vegetables for the children (Figure 2B). This change was made through the application of the nutritional education program. All students (*n* = 169) from grades 1 to 8 are offered school lunches rich in fresh FVs. No adverse side effects were reported by the participants during or after the intervention.

### 3.2. Participants’ Characteristics

There was a total of 101 eligible participants from grades 4 to 8, of which only 92 agreed to participate in the trial. All participants completed the pre-test questionnaire. After four months of nutrition education courses, all participants completed the post-test questionnaire. After excluding incomplete questionnaires, a total of 92 participants completed the nutrition knowledge questionnaire, 65 completed the attitude questionnaire, and 75 completed the behavior questionnaire (Figure 3).

The detailed distribution of participants by grade and gender is shown in Table 2. Overall, the total number of participants was 92, including 53 boys (58%) and 39 girls (42%), indicating that there were more boys than girls among the participants. Notably, the proportion of boys was highest in the fifth grade (80%) and the eighth grade (71%), while the proportion of girls was highest in the seventh grade (65%). Additionally, the grade with the highest number of participants was the sixth grade, with a total of 22 students (24%), whereas the grade with the lowest number of participants was the fourth grade, with only 16 students (17%). The students who agreed to participate in the trial showed a significant gender difference. Since the students who agreed to participate in this study showed a significant gender difference, a two-way repeated measures ANOVA will be used in the subsequent analysis to account for this difference.

### 3.3. The Improvement in Nutrition Knowledge After the Nutrition Education Course

The education of nutritional knowledge is divided into four parts: food category, choose healthy food, food servings, and health benefits of breakfast. After finishing the nutrition education course, the scores for all nutrition knowledge items improved compared to before the course. Among them, food categories, food servings, and the total score showed statistically significant differences (Table 3).

### 3.4. The Changes in Attitudes and Behaviors Toward the Consumption of FVs After the Nutrition Education Course

In promoting the consumption of FVs, in addition to increasing their supply in school lunches, the nutrition education course also taught students about different types of FVs, recommended the daily intake, and explained the health benefits of consuming them. Furthermore, three FVs cooking and tasting classes were organized to enhance student engagement. After completing the course, students showed a significant improvement in their attitude scores toward FVs consumption (*p* = 0.040) before adjusting for gender. However, after adjusting for gender, the difference was not statistically significant. While FV consumption behavior scores showed an upward trend, none of the changes were statistically significant (Table 4).

Further analysis was conducted to observe the effectiveness of the nutrition education across different grade levels. The results indicated that after completing the nutrition education course, fifth-grade students showed the greatest improvement in their nutrition knowledge scores, with an increase of 38.5%. The difference was statistically significant before adjusting for gender (*p* = 0.024); however, after adjusting for gender, the significance was no longer observed. Regarding attitude and behavior scores toward fruit and vegetable consumption, fourth-grade students showed the greatest improvement, with increases of 10.9% and 6.3%, respectively. Before adjusting for gender, this difference was statistically significant (*p*-values of 0.021 and 0.037, respectively). However, after adjusting for gender, the significance was no longer observed (Table 5).

### 3.5. Annual Consumption of FVs

The content of this nutrition education program was designed specifically for students in fourth grade and above. After its effectiveness was validated through this study, the school agreed to incorporate nutrition education as a regular part of the fourth-grade curriculum and continued to offer fresh FVs in the school lunches. The amount of annual FVs consumption was collected from the school’s order records, representing the total consumption of students from grades 1 to 8. We observed a steady increase, from 1853 pounds in 2017 to 6580 pounds in 2021, a 3.5-fold increase. The decrease in FVs consumption in 2020 was attributed to the COVID-19 pandemic, which caused students to stop attending in-person classes. (Figure 4). Although the school cannot distinguish the specific consumption by students in grades 4 to 8, the noticeable annual increase in consumption is estimated to be contributed by over 60% of students in grades 4 to 8.

## 4. Discussion

In 2010, Shin Kong Wu Ho-Su Memorial Hospital established the International Medical Cooperation Promotion Team: Shin Kong Medical Assistance Program to Palau (SKMP). This program dispatched medical personnel to Palau to address immediate medical needs, trained local healthcare workers, assisted with critical care referral services, and established a nutrition consultation room to promote healthy eating concepts. In 2014, TTM planned a farm-to-table project to grow seasonal organic FVs and provide them for free to school lunch programs. This initiative has already created an environment for children to increase their FVs intake. School lunches rich in FVs provide multiple benefits, including improving students’ nutritional knowledge, fostering healthier eating behaviors, and increasing FVs intake. The free meal program in Palau ensures that all students have access to fresh FVs. When combined with nutrition education and taste-testing activities, it further enhances students’ acceptance and intake of FVs. Based on constructs from Social Cognitive Theory, individual cognitive factors (such as nutritional knowledge) and the social environment (such as FV availability) are important factors in shaping healthy eating behaviors. We believe that if nutritional education is added, it will further enhance children’s healthy eating behaviors. This led to the initiation of our plan to implement nutrition education programs in public elementary schools in Palau.

Lippe et al. (2007) [25] conducted a Youth Risk Behavior System (YRBSS) to monitor six categories of priority health-risk behaviors in Pacific Island territories and found that only 19.8% of adolescents in Palau consumed at least five servings of vegetables and fruits daily, and only 8.9% consumed at least three servings of milk per day. In contrast, a high percentage of 34.9% of adolescents consumed at least one cup or can of soda or pop per day. These findings indicated poor dietary habits among adolescents [25]. Therefore, in addition to promoting the consumption of FVs, we also incorporated topics into the nutrition education curriculum such as understanding different types of food and their roles in the body, the importance of eating breakfast, and the negative health effects of sugary and high-fat foods. Previous research has shown that experiential learning activities can increase children’s willingness to eat unfamiliar vegetables [28]. Classroom nutrition education combined with FVs taste testing also demonstrated to improve children’s dietary intake [29]. According to Ehrenberg et al.’s research, repeated exposure to cooking activities not only increased participants’ contact with a variety of fruits and vegetables but also enhanced their tasting and acceptance of these foods. Specifically, involving children in hands-on processes, such as handling and preparing ingredients, boosted their interest in food and their likelihood of actively choosing healthy options. This provides empirical evidence supporting the improvement of children’s dietary behaviors, showing that such approaches are effective not only for short-term dietary preference changes but also for fostering long-term healthy eating habits [30]. Therefore, we decided to include not only lessons and mini-games to enhance the children’s interest in learning but also held cooking and tasting activities. Most students considered this course design to be novel and engaging. The present study also revealed similar results, showing a significant increase in students’ nutrition knowledge scores, and improvements in their attitudes towards consuming FVs. In the analysis by age group, we observed fourth-grade children had the lowest nutrition education scores among all age groups, and their scores after the course were also the lowest. We believe this may be due to the relative difficulty in understanding the questionnaire by children in this age group. In contrast, when compared to other age groups, their attitudes and behaviors toward the consumption of FVs showed greater improvements after completing the nutrition education course. This indicated that providing nutrition education at a younger age may lead to greater improvements in dietary attitudes and behaviors.

Several studies have demonstrated that multicomponent nutrition education programs can effectively improve knowledge, attitudes, and behaviors related to fruit and vegetable (FV) consumption among third-, fourth-, and fifth-grade students [31,32]. Hahnraths et al. combined classroom-based curricula with experiential learning strategies, such as cooking, to evaluate the short- and long-term effects on children’s FV intake. Following the course, the intervention group showed significant improvements in knowledge, taste preferences, and attitudes toward FVs. However, no significant differences were observed three months after the intervention [33]. These findings suggest that while nutrition education can produce short-term effects, long-term environmental support is essential for sustaining improvements in children’s dietary behaviors. In this study, after completing the 16-week course, students showed a significant improvement in their nutrition knowledge scores. However, their attitude and behavior scores toward FVs consumption only showed a slight increase. Nevertheless, through the regular implementation of annual nutrition education for newly promoted fourth-grade students and the long-term provision of fresh FVs in schools, we observed a gradual increase in FVs consumption over the years. This suggests that integrating nutrition education into core curricula—such as math, science, and literacy—along with establishing a supportive healthy eating environment, can contribute to the long-term improvement of FV intake.

This preliminary success included improvements in knowledge, attitudes, and behavior scores, as well as long-term observation showing an increase in fruit and vegetable consumption among elementary school students in Palau, which is the result of a combination of policy, systems, and environmental improvements (providing more fruits and vegetables), coupled with nutrition education. However, this study still has some limitations. First, we only observed the differences in students’ nutrition knowledge, attitudes, and behaviors before and after the nutrition education program, without a control group. The main reason for this is the small number of students in the participating school, where all students, including those not involved in the trial, received free FVs in their school lunches. We also lacked sufficient funding to identify a control group in other schools. This limitation is consistent with challenges highlighted in other educational intervention studies, such as the findings of Moscatelli et al. (2023), which provide additional context for understanding the lack of a control group in this study [34]. The research shows that providing healthy, scratch-cooked, and less processed lunches in schools also could increase children’s FVs consumption [35]. Therefore, the short-term nutrition education clearly improves children’s nutrition knowledge, but long-term provision of FVs in school lunches could promote children’s actual consumption of these foods in our study. Second, we emphasize the moderate consumption of seasonal and locally fresh fruits; therefore, the impact of the glycemic index (GI) of fruits on health has not been considered. Third, this study did not collect data on children’s anthropometrics, as the primary focus was on fruit and vegetable consumption and related attitudes. Fourth, several studies have also demonstrated that factors such as parental education level, the availability of FVs at home, and family income are important influences on children’s FVs intake [36,37]. However, in this study, we did not investigate these influencing factors. Thus, the inexistence of a control group and the lack of control of potential confounding variables that influence the dependent variables limit the validity of the present study.

## 5. Conclusions

Our pilot evaluation showed that implementing nutrition education and providing more fresh FVs in elementary school could improve the scores of nutritional knowledge, attitudes and behavior toward consuming FVs, particularly among fourth and fifth-grade students. Combined with the provision of FVs in schools, it can effectively increase children’s FVs intake. In the future, promoting this program to other schools will require not only the support of principals and teachers but also consideration of the adequacy of local kitchen facilities, the coordination of kitchen staff, and the suitability of their cooking skills.

## Figures and Tables

**Figure 1 nutrients-17-00994-f001:**
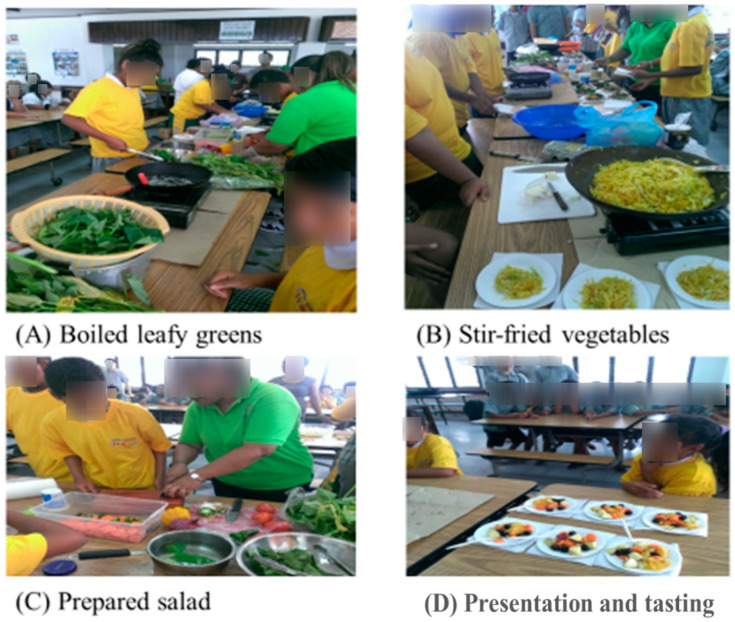
Hands-on cooking activities for nutrition education.

**Figure 2 nutrients-17-00994-f002:**
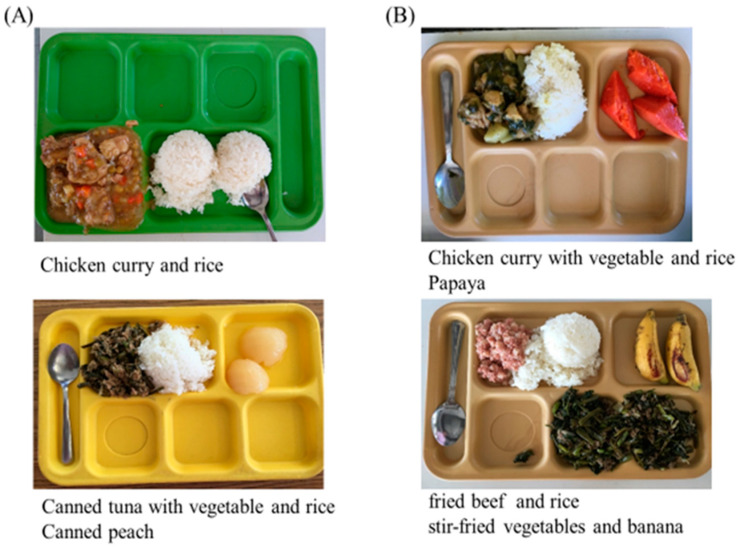
Examples of school lunches before (**A**) and after (**B**).

**Figure 3 nutrients-17-00994-f003:**
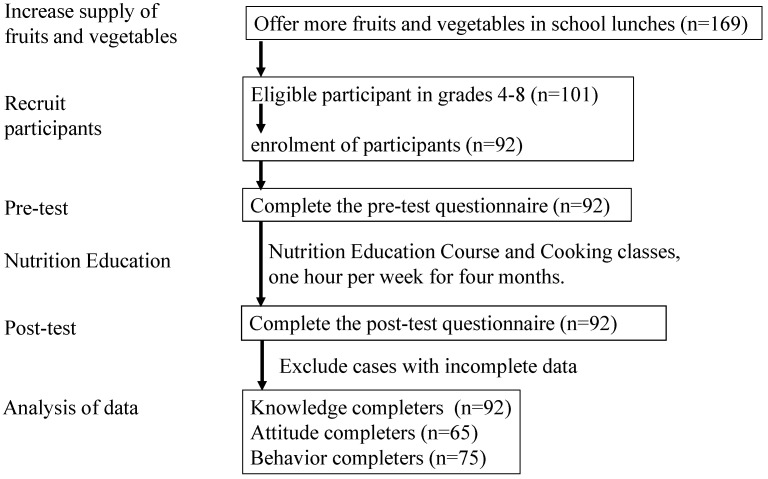
Flow diagram of participation.

**Figure 4 nutrients-17-00994-f004:**
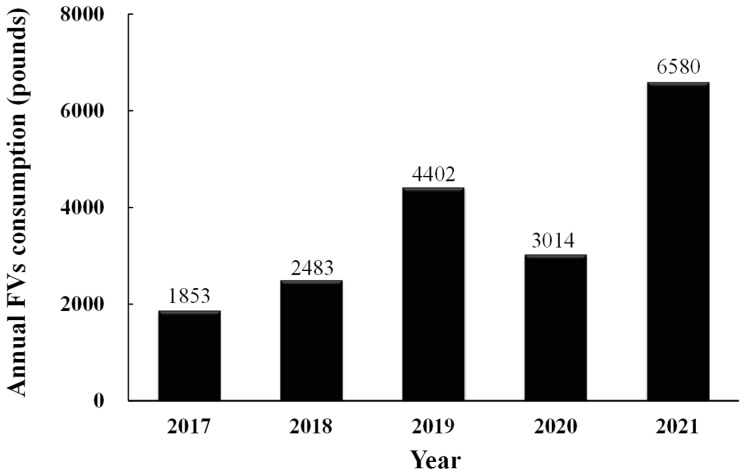
Annual FVs consumption from 2017 to 2021.

**Table 1 nutrients-17-00994-t001:** Summary of the Nutrition Education Program contents.

Nutrition Topics	Main Contents	Materials
Healthy eating	(A) Knowledge of nutrients (carbohydrate, protein, fat, vitamins, minerals, dietary fiber, and water) (B) Benefits of consuming FVs (C) Classroom Q&A (D) Group discussion	Electronic courseware Prize game
Food category	(A) Classification of foods (B) Nutritional value and recommended intake (C) Classroom Q&A (D) Mini-game	Electronic courseware Prize game
Food servings	(A) Recommended serving sizes provided by dietary guidelines (B) Examples of serving sizes for various foods (C) Mini-game (D) Classroom Q&A	Electronic courseware Model plates
Healthy snacks	(A) Negative health impacts of added sugars and trans fatty acids (B) Choose healthy snacks and restrict the intake of added sugars and trans fatty acids. (C) Read nutrition facts label (D) Group discussion (E) Classroom Q&A	Electronic courseware Visual cards
Cooking, learning, and testing of FVs	(A) How to choose fresh FVs (B) Food preparation (C) Learning simple cooking methods (boiled, stir-fried, and salad) (D) Tasting and sharing feedback	Cooking, learning, and testing of FVs

**Table 2 nutrients-17-00994-t002:** Demography of Participants.

Grade	Total *n* (%)	Boys *n* (%)	Girls *n* (%)	*X* ^2^	Freedom	*p* Value
4th	16 (17)	9 (56)	7 (44)	10.088	4	0.039
5th	20 (22)	16 (80)	4 (20)			
6th	22 (24)	10 (46)	12 (54)			
7th	17 (18)	6 (35)	11 (65)			
8th	17 (18)	12 (71)	5 (29)			

Data were presented as *n* (%). Differences between groups were tested using chi-square tests; *p* value < 0.05 was considered statistically significant.

**Table 3 nutrients-17-00994-t003:** Score of nutritional knowledge after nutrition education ^a^.

	Before (*n* = 92)	After (*n* = 92)	*p* Value
Food category	3.15 ± 1.99	3.83 ± 2.13	0.004 *
Choose healthy food	2.54 ± 0.87	2.67 ± 1.03	0.196
Food servings	0.42 ± 0.58	0.57 ± 0.70	0.003 *
Health benefits of breakfast	2.11 ± 0.99	2.21 ± 0.93	0.519
Total score	7.82 ± 3.89	8.72 ± 4.06	0.006 *

^a^ Data were presented as mean ± SD. * Differences between before and after nutrition education were tested using a two-way repeated measures ANOVA; *p* < 0.05 considered statistically significant.

**Table 4 nutrients-17-00994-t004:** Attitude and behavior scores of eating fruits and vegetables after nutrition education ^a^.

	*n*	Before	After	*p* Value
Attitude				
Eating fruits	65	6.87 ± 1.21	7.16 ± 0.87	0.343
Eating vegetables	65	6.34 ± 1.54	6.66 ± 1.42	0.813
Eating FVs	65	13.25 ± 2.45	13.85 ± 2.15	0.698
Behavior				
Eating fruits	75	8.69 ± 0.96	8.79 ± 0.72	0.872
Eating vegetables	75	7.65 ± 1.77	7.87 ± 2.18	0.753
Eating FVs	75	16.34 ± 2.24	16.70 ± 2.24	0.664

^a^ Data were presented as mean ± SD. Differences between before and after nutrition education were tested using a two-way repeated measures ANOVA; *p* < 0.05 considered statistically significant. FVs, fruits and vegetables.

**Table 5 nutrients-17-00994-t005:** Nutritional knowledge, attitudes and behavior of FVs-related scores of students at different grade levels before and after nutrition education ^a^.

	*n*	Before	After	*p* Value
Nutritional knowledge				
Grade 4th	16	6.62 ± 3.14	6.68 ± 4.42	0.050
Grade 5th	20	7.15 ± 4.75	9.90 ± 3.52	0.056
Grade 6th	22	8.77 ± 4.05	10.32 ± 4.13	0.675
Grade 7th	17	8.29 ± 3.68	8.82 ± 3.52	0.112
Grade 8th	17	8.00 ± 3.32	7.06 ± 6.70	0.795
Attitude of eating FVs				
Grade 4th	12	12.92 ± 2.53	14.33 ± 2.06	0.883
Grade 5th	12	12.83 ± 2.21	12.83 ± 2.37	0.126
Grade 6th	19	14.37 ± 1.95	14.53 ± 1.81	0.343
Grade 7th	12	12.78 ± 2.33	13.44 ± 1.59	0.659
Grade 8th	10	12.14 ± 3.53	13.43 ± 2.99	0.851
Behavior of eating FVs				
Grade 4th	14	16.86 ± 1.83	17.93 ± 0.27	0.536
Grade 5th	15	16.00 ± 2.29	16.53 ± 2.17	0.705
Grade 6th	22	16.5 ± 2.06	17.27 ± 1.80	0.053
Grade 7th	14	16.46 ± 1.98	16.31 ± 2.69	0.407
Grade 8th	10	15.29 ± 3.68	14.86 ± 3.02	0.293

^a^ Data were presented as mean ± SD. Differences between before and after nutrition education were tested using a two-way repeated measures ANOVA; *p* < 0.05 considered statistically significant. FVs, fruits and vegetables.

## Data Availability

The data supporting the reported results and conclusions can be found in the submitted figure and tables. Additional research materials and protocols that are relevant to the study are available from the corresponding author upon reasonable request.

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
