# Peer review of "A Pilot Evaluation of a School-Based Nutrition Education Program with Provision of Fruits and Vegetables to Improve Consumption Among School-Age Children in Palau"

_nutrients, 2025, doi:10.3390/nu17060994_

Round 1

Reviewer 1 Report

Comments and Suggestions for Authors

I read the article. The paragraph 53-60 is the same with 62-70. Please give details about the type of fruits, vegetables, quantity, reported to the age of children.

Calculated the glycemic index for the fruits.

Please describe the benefits of diet.

What was the weight of the children before the study? How is this modified? 

How many calories has the diet?

Do you have side effects after the meals?

Author Response

Response to Reviewer #1:

Comment 1: "The paragraph 53-60 is the same as 62-70."

Response: Thank you for pointing out this issue. We have removed the redundant content in lines 62-70 to ensure the manuscript is concise and clear.

Comment 2: "Please give details about the type of fruits, vegetables, quantity, reported to the age of children."

Response: The manuscript has been revised to include details about the types of fruits and vegetables, their respective quantities, and their distribution according to the age groups of children. These details have been added to the Methods section for better clarity (L98-L105).

Comment 3: "Calculated the glycemic index for the fruits."

Response: We emphasize the moderate consumption of seasonal and locally fresh fruits; therefore, the impact of the glycemic index (GI) of fruits on health has not been considered. Due to the lack of specific data required for this analysis, we regret that we are unable to calculate the GI of the fruits. This limitation has been clarified in the Discussion section (L346-L348).

Comment 4: "Please describe the benefits of diet."

Response: Thank you for your comment. We have expanded the discussion to highlight the benefits of the diet, including its role in improving nutritional knowledge and fostering healthier eating habits in children (L275-L279).

Comment 5: "What was the weight of the children before the study? How is this modified?"

Response: Thank you for your comment. This study did not collect data on children’s anthropometric as the primary focus was on fruit and vegetable consumption and related behaviors. This limitation has been acknowledged in the Discussion section (L348-L350).

Comment 6: "How many calories has the diet?"

Response: Thank you for your question. Lower-grade students are provided with 350-450 kilocalories per day, while middle- and upper-grade students are provided with 450-650 kilocalories per day. This information has been included in the Methods section (L106-L108).

Comment 7: "Do you have side effects after the meals?"

Response: Thank you for your inquiry. No adverse side effects were reported by the participants during or after the intervention. This has been clarified in the Results section (L177-L178).

Reviewer 2 Report

Comments and Suggestions for Authors

The purpose of this study is to investigate whether nutrition education (NE) can improve nutritional knowledge, eating behaviors, and increase the intake of FVs among school-age children. 

The manuscript is well structured and deals with a topic of great current interest and potential interest to the community. I have only a few minor suggestions for the authors.

Although the authors have highlighted the lack of a control group within the limits of the study, it would be appropriate to better explain these aspects by trying to justify them well. In this regard, the authors could take into consideration the following study:

Moscatelli et al., Assessment of Lifestyle, Eating Habits and the Effect of Nutritional Education among Undergraduate Students in Southern Italy, Nutrients, 2023, 15(13), 2894.

The study does not take into account external factors such as parental influence, socioeconomic status and the availability of organic food at home, which are well-documented determinants of children's eating behaviors. Also in this regard, the authors should try to find a rationale for this lack and explain it in the text.

Author Response

Response to Reviewer #2:

The purpose of this study is to investigate whether nutrition education (NE) can improve nutritional knowledge, eating behaviors, and increase the intake of FVs among school-age children.

The manuscript is well structured and deals with a topic of great current interest and potential interest to the community. I have only a few minor suggestions for the authors.

Comment 1: “Although the authors have highlighted the lack of a control group within the limits of the study, it would be appropriate to better explain these aspects by trying to justify them well. In this regard, the authors could take into consideration the following study:

Moscatelli et al., Assessment of Lifestyle, Eating Habits and the Effect of Nutritional Education among Undergraduate Students in Southern Italy, Nutrients, 2023, 15(13), 2894.”

Response: Thank you for pointing out this aspect and for suggesting a relevant reference. We have revised the manuscript to provide a more detailed justification for the lack of a control group. The small size of the school and the inclusion of all students in the intervention made it impractical to designate a separate control group. To strengthen the discussion, we have cited the study by Moscatelli et al. (2023) to draw parallels between our findings and similar challenges in conducting educational interventions without control groups (L338-L341).

Comment 2: "The study does not take into account external factors such as parental influence, socioeconomic status, and the availability of organic food at home, which are well-documented determinants of children's eating behaviors. Also in this regard, the authors should try to find a rationale for this lack and explain it in the text."

Response: We appreciate this valuable comment. We have further highlighted these literatures in the study limitations section, explained the reasons why this study could not include these external factors, and suggested that future studies should further explore the combined effects of these factors on children's eating behaviors (L353-L355).

Reviewer 3 Report

Comments and Suggestions for Authors

Dear Authors:

Regarding the manuscript with title “Evaluation of a school-based nutrition education program to improve fruit and vegetable consumption among school-age children”, I have major concerns. Also several minor comments were addressed.

Major Comments:

Comment 1: The inexistence of a control group and the lack of control of potential confounding variables that influence the dependente variables liomits the validity of the presente study

Comment 2: Lines 312-313. “attributed to the nutrition education program, but also to the long-term provision of a variety of FVs in school lunches”. How can authors diferentiate the relative contribution of each one of these factors? The increase of FVs consumption must be 100% attributed to the long-term provisiono f a variety of FVs in school lunches.

Comment 3: Sample size was calculated to ensure sufficient statistical power for the study's objectives?

Minor Comments:

Comment 1: Line 22: “and engage in regular physical activity”. I suggest authors to withdrawn this sentence, as the theme of the manuscript is related to Nutrition.

Comment 2: On line 29, authors refer that only students in grades 4-8 were invited to participate in the nutritional education program. On lines 31-32, authors present results from 1-8 grade. This sentence must be deleted.

Comment 3: Lines 32-34: “After completing the nutrition education course, students 32 in grades 4-8 showed improvements in their scores across all nutrition knowledge items compared 33 to before the course”. This is true, but only with statistically significance on Food category.

Comment 4: Line 33: Authors must refer the number of students from grades 4-8 that participated in the study.

Comment 5: Authors have to add a sentence regarding the significant improvements of nutritional knowledge on 5th grade.

Comment 6: Lines 47-52: Authors must check the grammar of the sentences presented on these lines.

Comment 7: Authors must delete lines 62-70 as i tis a repetition of lines 53-61.

Comment 8: On the subchapter of Study design, authors must refer the design of the present study.

Comment 9: Lines 115-116: “They then undergo a 4-month nutrition education program and activities, meeting once a week for one hour each session.” This totalize 16 sessions. On subchapter 2.2. authors only refer to 4 lessons and 3 cooking classes. I kindly ask authors to clarify this question.

Comment 10: Line 119: Authors must withdrawn the word “This”.

Comment 11: On subchapter 2.3. Questionnaire design, I suggest authors to add a Table with information regarding all questions presented on questionnaire (differentiating knowledge, attitudes and behaviors)

Comment 12: Lines 163. I suggest authors to change “2A” by “2B” and to refer that this change was made through the application of the nutritional education program

Comment 13: On Table 2, authors should make statistical analysis to averiguate potential significant differences between gender.

Comment 14: Authors have to change the name of subchapter 3.3. I tis related with nutritional knowledge and not with participants’ characteristics.

Comment 15: Line 229: “nutrition education as a regular part of the 4th-grade curriculum”. The anual FVs consumption presented on Figure 4 is regarding students from 4th to 8th grade, correct?

Comment 16: Line 318-320. I suggest authors to withdrawn this sentence, as i tis not related with the purpose of the study.

Comment 17: On Conclusions, authors must refer the results regarding attitudes and behaviors regarding fruit and vegatable consumption.

Author Response

Response to Reviewer #3:

Regarding the manuscript with title “Evaluation of a school-based nutrition education program to improve fruit and vegetable consumption among school-age children”, I have major concerns. Also several minor comments were addressed.

Major Comments:

Comment 1: The inexistence of a control group and the lack of control of potential confounding variables that influence the dependente variables liomits the validity of the presente study

Response: We agree with your point and will include this aspect in the limitations section of the study (L338-L341).

Comment 2: Lines 312-313. “attributed to the nutrition education program, but also to the long-term provision of a variety of FVs in school lunches”. How can authors diferentiate the relative contribution of each one of these factors? The increase of FVs consumption must be 100% attributed to the long-term provisiono f a variety of FVs in school lunches.

Response: Thank you for your comments. Indeed, we are unable to differentiate the relative contribution of each of these factors. We will revise the wording to be more conservative. “The short-term nutrition education clearly improves children's attitudes and behaviors towards consuming FVs, but long-term provision of FVs in school lunches could promote children's actual consumption of these foods (L343-L346).

Comment 3: Sample size was calculated to ensure sufficient statistical power for the study's objectives?

Response: We used the nutrition knowledge score as the data for the sample size analysis, setting the mean difference as 2.0, the standard deviation of differences as 4, the alpha two-sided as 0.05, and the power as 0.9. The sample size for a paired t-test was 44 participants, indicating that the sample size we collected has sufficient statistical power(L121-L125).

Minor Comments:

Comment 1: Line 22: “and engage in regular physical activity”. I suggest authors to withdrawn this sentence, as the theme of the manuscript is related to Nutrition.

Response: Thank you for your suggestion, this sentence has been deleted

Comment 2: On line 29, authors refer that only students in grades 4-8 were invited to participate in the nutritional education program. On lines 31-32, authors present results from 1-8 grade. This sentence must be deleted.

Response: Thank you for your suggestion, this sentence has been deleted.

Comment 3: Lines 32-34: “After completing the nutrition education course, students 32 in grades 4-8 showed improvements in their scores across all nutrition knowledge items compared 33 to before the course”. This is true, but only with statistically significance on Food category.

Response: Thank you for your suggestion, this sentence has been fixed as below.

Our results revealed that students in grades 4-8 (n=92) showed improved scores in all nutrition knowledge items after completing the nutrition education course compared to their performance before the course, but only with statistically significance on Food category (L29-L32).

Comment 4: Line 33: Authors must refer the number of students from grades 4-8 that participated in the study.

Response: Thank you for your suggestion, the number of students have been added. Please refer to the above sentence.

Comment 5: Authors have to add a sentence regarding the significant improvements of nutritional knowledge on 5th grade.

Response: Thank you for your suggestion, “that 5th-grade students showed a significant improvement in their nutrition knowledge scores, with an increase of 38.5%” have be added in L34-L35.

Comment 6: Lines 47-52: Authors must check the grammar of the sentences presented on these lines.

Thank you for your suggestion, we have checked the grammar and fixed it (L47-L52).

Comment 7: Authors must delete lines 62-70 as i tis a repetition of lines 53-61.

Response: Thank you for pointing out this issue. We have removed the redundant content in lines 62-70 to ensure the manuscript is concise and clear.

Comment 8: On the subchapter of Study design, authors must refer the design of the present study.

Response: Thank you for your suggestion, we added more details about study design (L98-L115).

Comment 9: Lines 115-116: “They then undergo a 4-month nutrition education program and activities, meeting once a week for one hour each session.” This totalize 16 sessions. On subchapter 2.2. authors only refer to 4 lessons and 3 cooking classes. I kindly ask authors to clarify this question.

Response: Thank you for your suggestion. It seems our choice of wording was not precise enough. We have made the necessary revisions. (L133-L134). Topic 1 was taught over 4 lessons, while topics 2, 3, and 4 were each covered in 3 lessons.

Comment 10: Line 119: Authors must withdrawn the word “This”.

Response: We have withdrawn the word “This” (L128).

Comment 11: On subchapter 2.3. Questionnaire design, I suggest authors to add a Table with information regarding all questions presented on questionnaire (differentiating knowledge, attitudes and behaviors)

Response: Thank you for your suggestion. We put the table in the supplementary material (Table S1).

Comment 12: Lines 163. I suggest authors to change “2A” by “2B” and to refer that this change was made through the application of the nutritional education program

Response: Thank you for your suggestion, we changed “2A” by “2B” and added the above instructions (L175-L176).

Comment 13: On Table 2, authors should make statistical analysis to averiguate potential significant differences between gender.

Response: Thank you for your suggestion, we added p value by using chi-square tests in table 2.

Comment 14: Authors have to change the name of subchapter 3.3. I tis related with nutritional knowledge and not with participants’ characteristics.

Response: Thank you for your suggestion, we have revised it to: "The improvement in nutrition knowledge after the nutrition education course." (L205).

Comment 15: Line 229: “nutrition education as a regular part of the 4th-grade curriculum”. The anual FVs consumption presented on Figure 4 is regarding students from 4th to 8th grade, correct?

Response: The content of this nutrition education program was designed specifically for students in 4th grade and above. After its effectiveness was validated through this study, the school agreed to incorporate nutrition education as a regular part of the 4th-grade curriculum and continued to offer fresh FVs in the school lunches. The amount of annual FVs consumption was collected from the school's procurement records, representing the total consumption of students from grades 1 to 8. We observed a steady increase, from 1,853 pounds in 2017 to 6,580 pounds in 2021, a 3.5-fold increase. The decrease in FVs consumption in 2020 was attributed to the COVID-19 pandemic, which caused students to stop attending in-person classes. (Fig 4). Although the school cannot distinguish the specific consumption by students in grades 4 to 8, the noticeable annual increase in consumption is estimated to be contributed by over 60% of students in grades 4 to 8. (L247-257).

Comment 16: Line 318-320. I suggest authors to withdrawn this sentence, as i tis not related with the purpose of the study.

Response: Thank you for your suggestion, we have withdrawn this sentence.

Comment 17: On Conclusions, authors must refer the results regarding attitudes and behaviors regarding fruit and vegetable consumption.

Response: We have revised in line 357-359 based on your suggestion.

Round 2

Reviewer 1 Report

Comments and Suggestions for Authors

Dear authors, the article is very interesting and right now have a lot of informations about the children s meal. We found the type of vegetables and fruits that children eat at school, the quantity, the preference, etc. I think now the article is complete and show us all the aspects of the children meal in school.

Author Response

Comment: "The article is very interesting and now contains a lot of information about children’s meals, including the types, quantity, and preferences of fruits and vegetables. I think the article is now complete and covers all aspects of school meals."

Response:
Thank you for your positive feedback. We are pleased that you find the article comprehensive and informative. Your support is greatly appreciated. Please let us know if any further refinements are needed.

Reviewer 3 Report

Comments and Suggestions for Authors

My comments were fully addressed by authors.

Author Response

Comment: "My comments were fully addressed by authors."

Response: Thank you for your confirmation and feedback. We appreciate your time and effort in reviewing our manuscript.